# Nano-Confined Tin Oxide in Carbon Nanotube Electrodes via Electrostatic Spray Deposition for Lithium-Ion Batteries

**DOI:** 10.3390/ma15249086

**Published:** 2022-12-19

**Authors:** Alexandra Henriques, Amin Rabiei Baboukani, Borzooye Jafarizadeh, Azmal Huda Chowdhury, Chunlei Wang

**Affiliations:** 1Department of Mechanical and Materials Engineering, Florida International University, Miami, FL 33174, USA; 2Center for the Study of Matter at Extreme Conditions (CeSMEC), Florida International University, Miami, FL 33199, USA

**Keywords:** lithium-ion battery, tin oxide, carbon nanotube, electrostatic spray deposition

## Abstract

The development of novel materials is essential for the next generation of electric vehicles and portable devices. Tin oxide (SnO_2_), with its relatively high theoretical capacity, has been considered as a promising anode material for applications in energy storage devices. However, the SnO_2_ anode material suffers from poor conductivity and huge volume expansion during charge/discharge cycles. In this study, we evaluated an approach to control the conductivity and volume change of SnO_2_ through a controllable and effective method by confining different percentages of SnO_2_ nanoparticles into carbon nanotubes (CNTs). The binder-free confined SnO_2_ in CNT composite was deposited via an electrostatic spray deposition technique. The morphology of the synthesized and deposited composite was evaluated by scanning electron microscopy and high-resolution transmission electron spectroscopy. The binder-free 20% confined SnO_2_ in CNT anode delivered a high reversible capacity of 770.6 mAh g^−1^. The specific capacity of the anode increased to 1069.7 mAh g^−1^ after 200 cycles, owing to the electrochemical milling effect. The delivered specific capacity after 200 cycles shows that developed novel anode material is suitable for lithium-ion batteries (LIBs).

## 1. Introduction

Continuous advancement in lithium-ion battery (LIB) technology is crucial for modern life, which relies on the widespread integration and use of portable and wearable devices [1,2,3,4,5]. While recent research has introduced novel anode materials, including carbon-based anodes [6,7,8,9], alloy-based anodes [10,11,12,13], and even ceramics [14,15,16,17,18,19,20], currently the majority of lithium ion batteries still use graphite as an anode material. Compared to graphite, a typical intercalation anode with a low specific capacity, metal oxides (MOs) with an alloying reaction mechanism and/or conversion have attracted a lot of attention as a possible anode material for LIBs in recent years due to their high theoretical capacities, low costs, and high density as active anode materials [21]. Among different types of MOs, tin oxide (SnO_2_) has been widely suggested for application in LIBs due to its high theoretical capacity (1494 mAh g^−1^) and low cost [22,23]. Moreover, SnO_2_-based anode materials can be paired with different cathode materials due to the possibility of using these anodes at a relatively low potential [24]. Two-step lithiation processes for SnO_2_ anodes have been suggested:SnO_2_ + 4 Li^+^ + 4 e^−^ → Sn + 2 Li_2_O (1)
Sn + x Li^+^ + x e^−^ ↔ Li_x_Sn (0 ≤ x ≤ 4.4) (2)

However, performance with SnO_2_ was limited due to low conductivity, large volume expansion (~300%) leading to mechanical instability, and unstable solid electrolyte interfaces (SEIs) during cycling [25]. Recently, nanostructuring of various metals and metal oxides to alleviate volume expansion during charging has been reported, including the use of titanium, titanium oxide, and zinc oxide [12,19]. Following a similar approach, nanostructuring the particle size of SnO_2_ and compositing SnO_2_ with carbonaceous materials has been studied to overcome these problems and enhance the electrochemical performance of SnO_2_ [21,26,27]. Until now, different types of carbon materials, such as graphene, carbon nanofibers (CNFs), and carbon nanotubes (CNTs), have been incorporated with SnO_2_ [28,29,30]. CNT-based anode materials were widely applied in LIBs due to their excellent electrical conductivity, chemical stability, and mechanical properties [31]. The confinement of nanostructured energy materials is an additional method of improving the performance of LIBs [32,33].

To date, different types of SnO_2_/CNTs anode materials with enhanced electrochemical properties have been evaluated for energy storage applications [28,34]. Nevertheless, compositing CNTs with SnO_2_ by depositing the metal oxide on the wall of the CNT leads to poor life cycles and lithium storage performance due to severe agglomeration [21,35]. For example, Cheng et al. provided SnO_2_-CNTs composites through the direct growth of SnO_2_ on the wall of CNTs. The electrochemical results show weak stability for the anode material during charge/discharge cycles [36]. The initial discharge capacity obtained was 1708 mA hg^−1^, which was reduced to 546 mA hg^−1^ after 100 cycles at a current density of 50 mA g^−1^. Very recently, Cheng et al. developed a core–shell structured C@SnO_2_@CNTs composite. The fabricated anode delivered a reversible capacity of 850 mAh g^−1^ at a current density of 200 mA g^−1^ [37].

Electrostatic spray deposition (ESD) is a facile deposition of thin films or thick coatings on a variety of substrates, which applies a voltage difference between a solution source and a substrate that is heated. This causes the solution to be atomized and deposited uniformly on the aforementioned conductive substrate with different morphologies [38,39,40,41,42]. There are many advantages of using 3D materials as the current collector, such as a short diffusion length for Li ions, high electronic conductivity, the ability to suppress the growth of Li dendrite, and a large surface area which in turn can help to increase the mass of the active material. In this regard, three-dimensional (3D) nickel foams have been advantageously utilized as electrode substrates as they offer a large active surface area and a highly conductive continuous porous 3D network. The benefit of using metal foam as a current collector in Li-ion batteries is that the redox reaction occurs under improved conditions at the junction of the metal frame, the active material, and the electrolyte [43,44]. Herein, we propose a facile binder-free strategy to prepare a confined SnO_2_ nanoparticle inside multiwall CNTs (MWCNTs) with various weight ratios through the ESD method for LIBs. This unique structure could not only buffer the volume change in SnO_2_ during cycling processes but also increase its conductivity, leading to excellent electrochemical performance for LIBs. The developed anode material exhibited the specific capacity of 770.6 mAh g^−1^ at 100 mAg^−1^ at 100 cycles. This study demonstrates that confined SnO_2_ in CNTs via the ESD method is a promising anode electrode for LIBs.

## 2. Experimental Section

### 2.1. Material Synthesis

MWCNTs (outer diameter 8–15 nm, inner diameter 3–5 nm, and length of 10–50 μm, Cheap Tubes Inc., Grafton, VT, USA) were, initially, opened up and then cut into 0.2–1 μm long segments before being refluxed in HNO_3_ (70 wt.%) at 140 °C for 14 h (the metal catalyst residues were also removed during this process). After dissolving the tin (IV) acetate ethanol (50% *w*/*w*), the CNTs were immersed into an aqueous solution attaining a desired weight ratio of SnO_2_ to CNT (either 20%, 25%, or 30%). Utilizing capillary forces associated with CNTs, tin (IV) acetate was introduced into the inner walls with stirring and subsequent ultrasonic liquid mixing for 3 h. After slowly drying the mixture, to oxidize the tin (IV) acetate into SnO_2_ in the mixture, it underwent heat treatment at 280 ºC for 3 h. In order to identify the confined vs. unconfined SnO_2_, the control group samples were denoted as unconfined SnO_2_ on CNTs. The confined SnO_2_ in films was synthesized through the ESD method. For preparing the precursor solution, CNTs and SnO_2_ were dissolved in 25 mL of a solvent mixture of 1,2-propanediol prior to ESD. After fixing the distance between the metal nozzle and substrate at 3 cm, a DC voltage of 5–7 kV was applied. The flow rate of the precursor was set at 3 mL h^−1^ and the films were prepared on Ni foam at 250 °C for 2 h. The standard procedure for mitigation of experimental error has been followed to ensure reproducibility. Each procedure has been conducted several times, and the average results have been reported in the following sections.

### 2.2. Materials and Electrochemical Characterizations

The microstructure of the samples was characterized by field-emission scanning electron microscopy (FESEM, JEOL 6335, JEOL, Tokyo, Japan) and high-resolution transmission electron microscopy (HRTEM, Phillips CM-200, Philips, Amsterdam, The Netherlands). The X-ray diffraction (XRD) analysis was conducted via Cu Kα radiation (SIEMENS-D5000, Siemens, Munich, Germany). Electrochemical tests were carried out using CR2032 coin-type cells assembled in an argon-filled glove box (VAC Nexus I, Vacuum Atmospheres Co., Hawthorne, CA, USA). The SnO_2_/CNT films on the nickel foams were used as working electrodes, while lithium was used for counter and reference electrodes. Celgard 2400 (Celgard, Charlotte, NC, USA) and 1 M lithium hexafluorophosphate (LiPF_6_) in ethylene carbonate—diethyl carbonate (1:1, volume ratio) were used as the separator and electrolyte, respectively. Cyclic voltammetry tests were carried out at 0.2 mVs^−1^ using a Verstile Multichannel Potentiostat (VMP3, Bio-Logic, Seyssinet-Pariset, France). The galvanostatic charge/discharge tests were performed in the voltage range of 0.01–3 V through a NEWARE BTS-610 Battery Test System (Neware, Shenzhen, China).

## 3. Results and Discussion

Figure 1 shows the schematic of the unconfined and confined SnO_2_ in CNT samples. The resulting XRD diffraction patterns of the synthesized materials are shown in Figure 2. The XRD patterns of samples of SnO_2_, confined SnO_2_ in CNTs, and unconfined SnO_2_ on CNTs show four major diffraction peaks at the (110), (101), (200), and (211) planes of the tetragonal rutile SnO_2_ phase (JCPDS card No. 41-1445) [45]. The XRD results from the sample of SnO_2_ alone confirmed that the tin (IV) acetate precursor was indeed oxidized as desired. The XRD pattern for the MWCNTs was in accordance with XRDs conducted on CNTs in the published literature [46,47]. The results show minimal difference between the samples of confined SnO_2_ in CNTs and unconfined SnO_2_ on CNTs [48]. Owing to the broad (110) peak of SnO_2_, it not possible to discriminate the characteristic peak corresponding to the (002) planes from the graphene matrix [32].

ESD is a valuable technique for realizing porous film deposition and achieving a large electrode material surface area [44]. The morphology and structure of the as-prepared 20% confined SnO_2_ in CNT samples obtained through ESD were examined by SEM at different magnifications (Figure 3). Figure 3a–c shows that the deposited material on the Ni foam substrate has a porous morphology. Figure 3d shows the well-dispersed CNTs, and the confinement of SnO_2_ particles is further supported by the fact that there is no external agglomerated SnO_2_ observed whose morphology is similar to other ESD-synthesized metal oxide materials [49,50,51].

Figure 4a–d shows the TEM images of the 20% confined SnO_2_ in CNTs composites. High-resolution TEM images, shown in Figure 4a,b, show the wall structure of MWCNTs at different magnifications. The darker spots within the CNTs are the SnO_2_ nanoparticles with a typical one-dimensional morphology that are well confined inside the MWCNTs. In Figure 4c,d, more carbon nanotubes and SnO_2_ particles are shown. They are without visible SnO_2_ nanoparticles outside of the tube, as shown in Figure 4c,d. As seen in the TEM images, it is clear that the SnO_2_ particles are not closely packed inside the MWCNTs. This unique structure prepares enough buffer space for the huge change in SnO_2_ volume that occurs during cycling of LIBs.

The electrochemical performance of the samples is examined by galvanostatic discharge–charge measurements and cyclic voltammetry (CV). Figure 5 shows the CV curves for various samples in their third cycle with a scan rate of 0.2 mV s^−1^. The weak irreversible cathodic peak at around 0.76 V is related to SEI formation [52]. Figure 5 shows the characteristic oxidation and reduction peaks for the tin oxide only control sample. The first reaction step involves alloying of metallic Sn with Li ions that lay between the 0.1 and 0.5 V peaks [53]. The CV results for pure SnO_2_ are shown in Appendix A. SEI formation corresponds to the peak at around 1.0 V, and reduction of SnO_2_ to metallic Sn is attributed to the peak at 1.35 V in the first reaction. The reverse reaction that corresponds to the peak at 1.2 V can exclusively be seen in the pure SnO_2_ and 20% or 30% confined SnO_2_ in CNT samples. The peak at around 0.55 V shows that the Li de-alloys from the Li_x_Sn compound. CNTs alone show very small current response. Similarly, in contrast to the confined sample, samples with unconfined SnO_2_ do not show a noticeable redox peak (Appendix A). The current response for SnO_2_ was large; however, the response deteriorates rapidly with increasing cycles, leading to increased half-cell instability over time.

Galvanostatic charge and discharge of half-cell samples with 20% confined SnO_2_ in CNT active material at the rate of 100 mAhg^−1^ were cycled for the 1st, 2nd, 10th, 50th, 100th, and 150th cycles, as shown in Figure 6. The first discharge and charge profiles are in accordance with the cyclic CV results. Lithium storage performance in the first discharge and charge was 1301 mAhg^−1^ and 641 mAhg^−1^, respectively. However, a sharp drop was detected in subsequent cycles, until cycle 10, due to SEI formation, the irreversible reaction of SnO_2_ to Sn, and amorphous Li2O. For cycle 40 to 150, the specific capacities continually increased to around 915 mAhg^−1^. Such trends can be explained by the electrochemical milling effect sometimes observed in anodes with metal oxide active materials, where the size of the particles reduces with cycling of the lithium ion batteries [54]. Since the metal oxide’s surface area increases due to the decrease in particle size, an increased surface is available for reaction between the Li^+^ ions and metal oxide particles.

The cycle performance of the samples is presented in Figure 7. The specific capacity decreases during the first 20 cycles for all samples. However, the samples with confined SnO_2_ in CNTs (20, 25, and 30%) could recover a portion of the capacity by cycle 40, and this trend continued through the to the 100th cycle. This trend is due to the electrochemical milling process, as well as to the enhanced reversibility coming from the first reaction step involving metallic Sn and Li_2_O formation from SnO_2_ and Li^+^ [55]. Although the initial performance of the unconfined sample was good, the capacity continued to decrease after further cycling and, unlike confined samples, no capacity recovery was observed. Previous study shows that, for nanoscale SnO_2_ anodes, the first reaction step can achieve greater reversibility, providing a theoretical capacity of 1490 mAh g^−1^. As SnO_2_ and metallic Sn in the anode materials go through electrochemical milling, this causes a reduction in particulate size and reduces the exposure of the unreacted surface, which allows for the enhancement to specific capacity. The sample with only a CNT had a low specific capacity as expected; however, this sample showed noticeable stability [56,57]. After 20 cycles, the CNT alone, 20% SnO_2_ outside of a CNT, 30% SnO_2_ inside a CNT, 25% SnO_2_ inside a CNT, and 20% SnO_2_ inside a CNT delivered a specific capacity of 183.5, 491.4, 523.5, 390.4, and 558.3 mA hg^−1^, respectively. The cyclic performance of the 20% confined SnO_2_ inside a CNT at 100 mAhg^−1^ up to 200 cycles is shown in Appendix A. It can be clearly seen that the specific capacity decreases during the first 15 cycles and then subsequently increases throughout the next 200 cycles. The specific capacities at cycles 1, 2, 100, 150, and 200 are 1312.3, 686.5, 770.6, 943.1, and 1069.7 mAh g^−1^, respectively. The 20% confined SnO_2_ inside a CNT sample showed the best specific capacity of 770.6 mAh g^−1^ after 100 cycles, which is comparable to previous CNT/SnO_2_-Sn anode results for LIBs [58,59,60]. The synergy between confined SnO_2_ and CNT composites could be attributed to interfacial characteristics, which can determine the bond ratios of Sn-C and Sn-O-C, where the former would provide superior and more facile electron transfer than the latter (in addition to structural stability) [36]. Such detailed analyses are of interest for future studies. The results show that obtaining confined 20% SnO_2_ inside a CNT using the ESD technique is an effective approach for increasing conductivity and controlling volume changes in anode materials during cycling. Future studies are required in a full-cell setup to evaluate the electrochemical performance of hybrid anodes for practical applications [61,62].

Figure 8a,b show the EIS characterization of the CNT, unconfined SnO_2_ on a CNT, and confined SnO_2_ in a CNT anodes before and after cycling. As shown, each Nyqiust curve consists of semicircles and a straight line at high to low frequency, respectively. The Nyquist curves are simulated using a modified Randles equivalent circuit (inset of Figure 8a). The elements in the circuit are solution resistance (R1), charge transfer resistance (R2), constant phase element (CPE), and Warburg diffusion (W). Solution resistance for all the samples was in a similar range (from 11 to 15 Ω). The semicircle diameter of the Nyquist curve shows the charge transfer resistance. For fresh cells, the charge transfer resistance of the CNT anode (712.8 Ω) is much higher than the unconfined SnO_2_ (372.1 Ω) and confined SnO_2_ (250.8 Ω) CNT anodes, which indicates the better electronic conductivity and kinetics of CNT samples with SnO_2_. After 100 cycles, the confined SnO_2_ in a CNT electrode showed very low charge transfer resistance of 67.9 Ω compared to a fresh cell, indicating improved charge transfer kinetics after cycling [39].

## 4. Conclusions

In summary, we propose a binder-free anode material obtained using the ESD method and based on a nano-confined SnO_2_ in CNT composite for lithium storage applications. The as-prepared 20% confined SnO_2_ in a CNT anode revealed a high reversible specific capacity of 770.6 mAh g^−1^, and the capacity of the anode increased when cycling up to 150 cycles. The electrochemical results show that 20% confined SnO_2_ can improve structural stability and increase the conductivity of composites while decreasing the diffusion pathway of Li ions when applied in LIBs.

## Figures and Tables

**Figure 1 materials-15-09086-f001:**
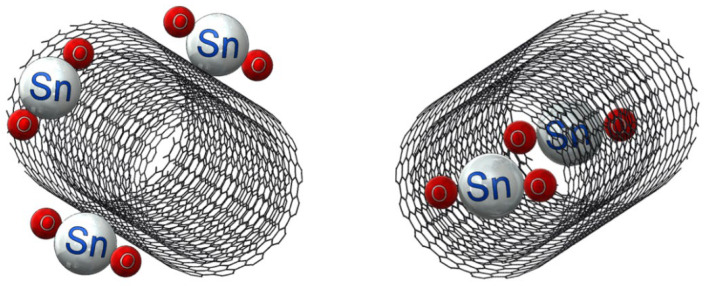
Schematic of unconfined SnO_2_ on a CNT and confined SnO_2_ in a CNT.

**Figure 2 materials-15-09086-f002:**
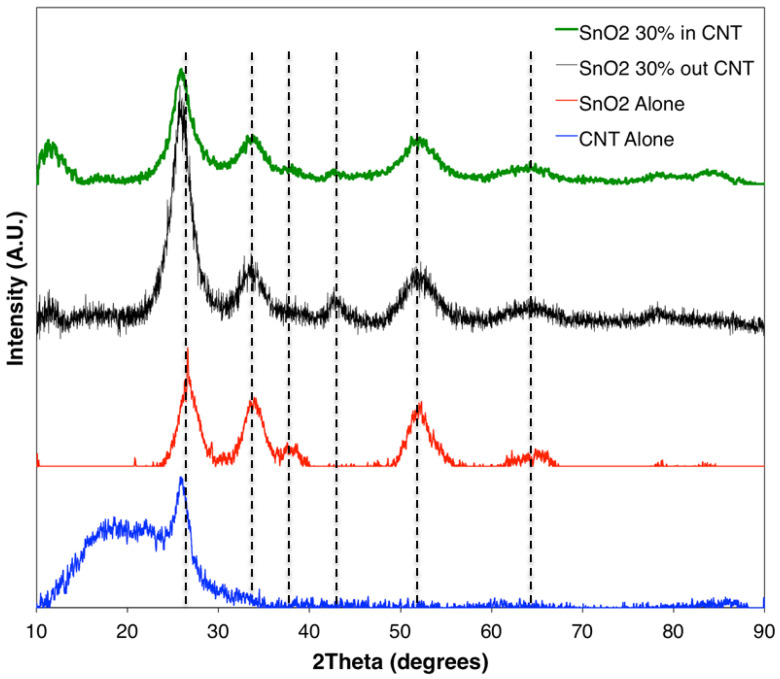
XRD plots for samples with confined and unconfined SnO_2_ in/on a CNT, and also patterns for pure SnO_2_ and a CNT.

**Figure 3 materials-15-09086-f003:**
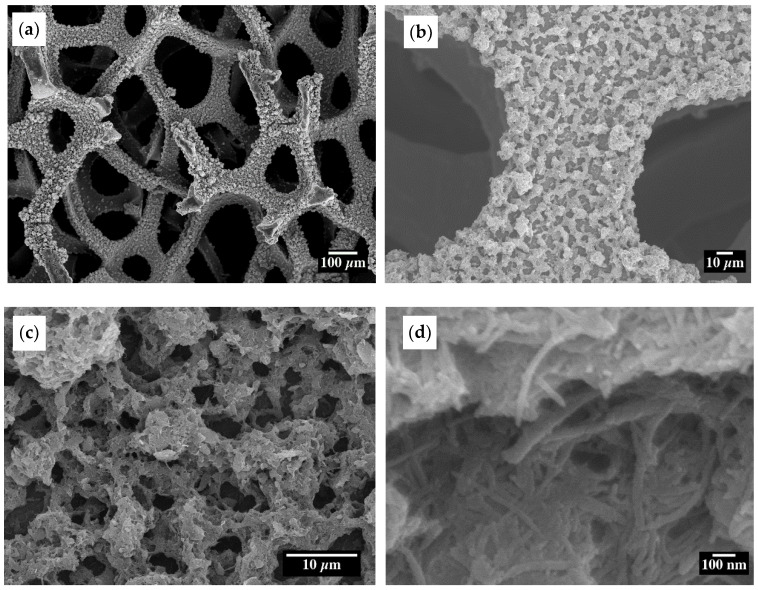
SEM images of the confined SnO_2_ in a CNT at different magnifications (**a**–**c**) uniform porous morphology, and (**d**) dispersed CNT.

**Figure 4 materials-15-09086-f004:**
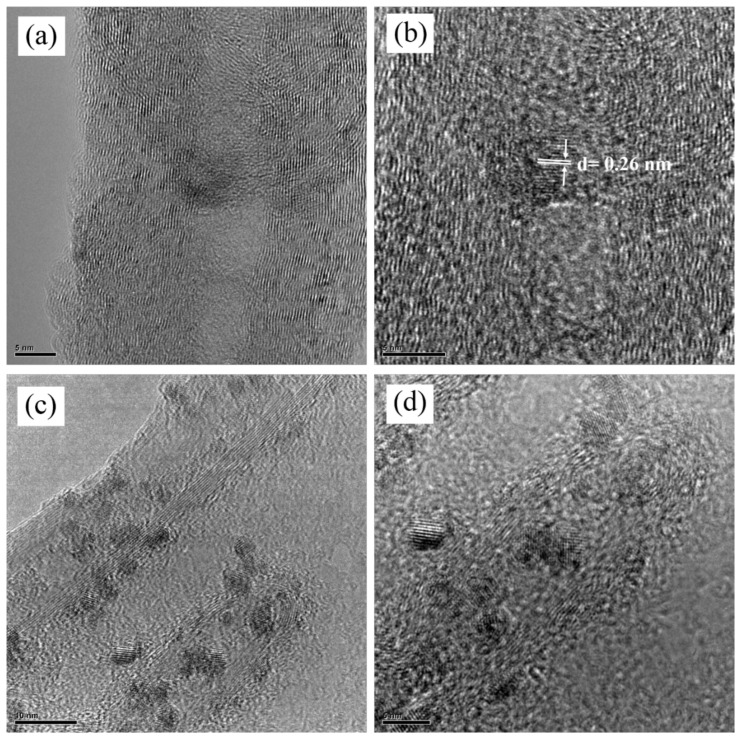
(**a**,**b**) TEM image of an individual vertically oriented carbon nanotube with a fully confined tin oxide nanoparticle. The atomic plane’s d spacing confirms the identification of the nanoparticle, and (**c**,**d**) confirm that SnO_2_ particles are confined inside the CNTs.

**Figure 5 materials-15-09086-f005:**
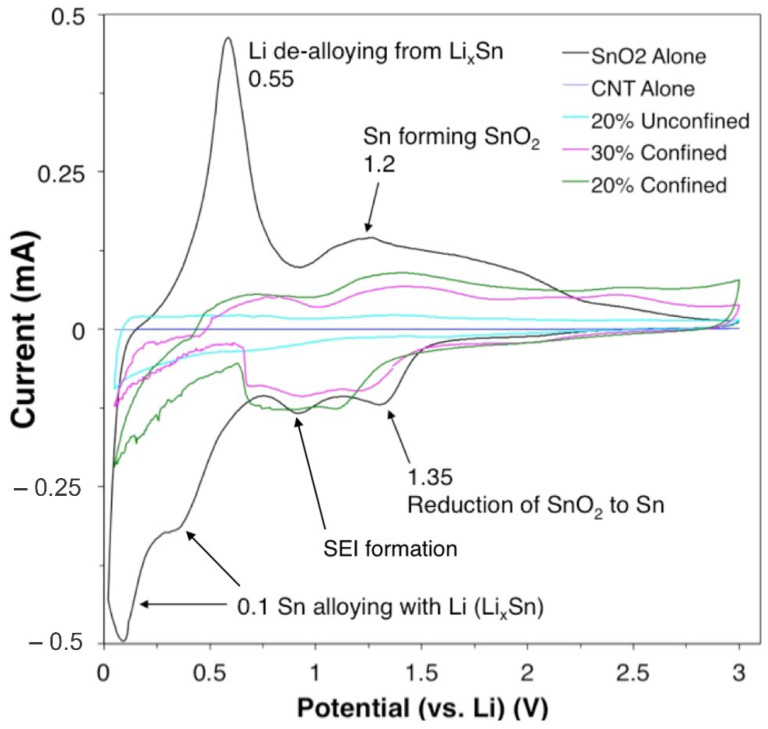
CV curves for control group and experimental samples.

**Figure 6 materials-15-09086-f006:**
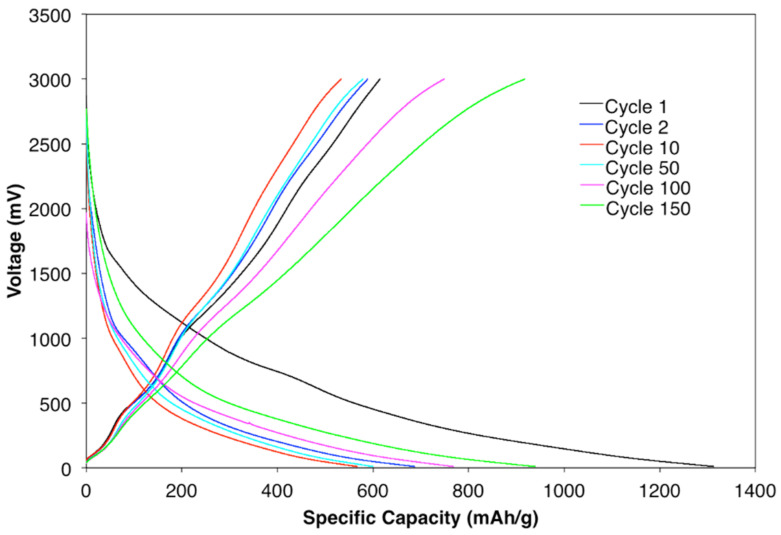
Galvanostatic charge/discharge curves for a sample with 20% confined SnO_2_ in a CNT.

**Figure 7 materials-15-09086-f007:**
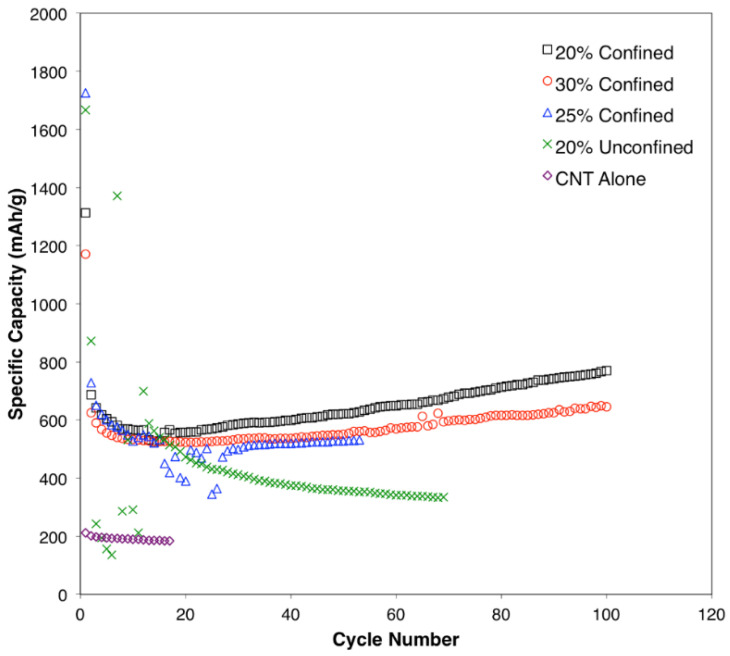
Cycle performance for the synthesized anode materials.

**Figure 8 materials-15-09086-f008:**
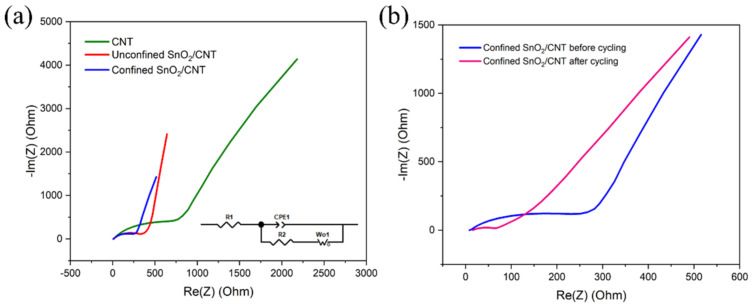
Electrochemical impedance (**a**,**b**) characterization of CNT anodes and anodes with unconfined SnO_2_ and confined SnO_2_ inside a CNT before and after 100 cycles (the inset shows the equivalent circuit).

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
