# Peer review of "Nano-Confined Tin Oxide in Carbon Nanotube Electrodes via Electrostatic Spray Deposition for Lithium-Ion Batteries"

_materials, 2022, doi:10.3390/ma15249086_

Round 1

Reviewer 1 Report

In this study, authors synthesized a high electrochemical performance electrode material with a confined SnO2 nanoparticle inside multiwall CNTs with various weight ratios through the ESD method for LIBs. This unique structure could not only buffer the volume change of SnO2 during cycling processes but also increase its conductivity, leading to excellent electrochemical performance for LIBs. The work is interesting and has a certain potential. However, the article would need a major revision, as reading it can be quit confusing at times (more on this below). If the following problems can be corrected, I think it would be more suitable for publication in this journal.

1:In the text, there are a number of inaccuracies, and it is confusing in superscript and subscript, such as “SnO2, 1494 mA hg-1, Li+, HNO3, 3 mL h−1”.

2: In the Materials synthesis section, the author claims that “the control group samples are denoted as SnO2-out-CNT”. In my opinion, the material Characterizations (SEM) and electrochemical performance of SnO2-out-CNT should be tested for more understanding the profit modification.

3: The paper indicated that CNTs were prepared by cutting MWCNTs ,but SEM images of the SnO2-in-CNT were described layered wall structure of MWCNTs in different magnifications in Figure 3 (a-d) . It is difficult to understand the logic of your argument.

4: The Schematic illustration of SnO2-out-CNT and SnO2-in-CNT materials should be added

5: I recommend adding Electrochemical impedance spectroscopy (EIS) measurement of CNT Alone, SnO2 Alone, SnO2-out-CNT and SnO2-in-CNT

6:Some recent work are suggested to be cited to enhance the background such as Journal of Energy Chemistry 72 , 2022, 143-148; Rare Metals.2021;40(12):3494―3500; Advanced Powder Materials, 2022, 1, 100057.

Author Response

Dear reviewer, 

We thank you for your insightful review. We tried to address your concerns to the best of our ability. We hope that the answers below are satisfactory to your inquiry.

Reviewer 2 Report

The binder-free anode material for lithium storage applications using the ESD approach based on nano-confined SnO2 in CNT composite was presented in the publication. Compared to other composites, the 20% SnO2-in-CNT as manufactured performed exceptionally well. The results of this exciting article were good, however, the manuscript needs significant reworking before it can be accepted.

line 58 - 60: Rephrase the statement to make scientific sense

Line 62 -63: Provide exact figures of the capacity retention and number of cycles.

Line 67: on a variety of substrates. use coma, not full stops

Line 93: The motivation for the use of Nickel foam as a substrate should have been provided in the introduction.

Line 101: Give description; is the working electrode anode or cathode in your introduction you mention SnO2 is your anode.

The Manuscript should also include results of EIS( electrochemical impedance spectroscopy) for conductivity and charge tranfers resistance comparisons of the different synthesized composite.

Line 121 - 128: Give more information on the different morphologies observed on the SEM images

Line 135: "The darker spot within CNTS is the SnO2 nanoparticles" use TEM-EDX to support this statement, due to that EDX can be used for elemental composition.

Line 161 -162 What is the reason for a higher current response for SnO2 compared to other CV Curves provided?

Author Response

Dear Reviewer,

We thank you for your insightful review. We tried to address your concerns to the best of our ability in a given short time. We hope that the answers below are satisfactory to your inquiry.

Reviewer 3 Report

This manuscript proposes nano-confined tin oxide in carbon nanotube electrodes via electrostatic spray deposition for lithium-ion batteries. The topic is interesting, and certainly consistent with the contents to be proposed to the readers of “Materials”. However, the manuscript is not so well written and should be improved to be read with pleasure: this represents an important aspect in the current scenario of publications in international journals. Overall, I think that this manuscript could be accepted if the Authors will be able to take into account the following major revisions (in terms of bibliographic updates, grammar corrections and content deepening):

-        Detailed revisions: I spent several hours reading this manuscript, and Authors are asked to follow carefully the attached PDF file where I highlighted some points to be addressed. The attached file also contains language mistakes and typos (they are many in this work and should not be present when submitting a manuscript to an international journal: Authors are asked to check the manuscript better next time); some questions related to manuscript contents could also be present and Authors must consider them properly before submitting the revised manuscript. A point-by-point reply is required when the revised files are submitted.

-        Considering the amount of mistakes and typos present in this manuscript, a further check carried out by a native English speaker or by a professional English language center is suggested.

-        The Introduction should give a wider overview on the present scenario related to new oxide electrodes for batteries, both in terms of recently published reviews and research articles. In particular, newly designed materials for Li and post-Li batteries are missing and a paragraph on this topic is highly suggested to be added in the Introduction. Authors are invited to go through the literature published in the last six months on these issues, and also on concepts developed some years ago in this field. Some of them are also mentioned in the above mentioned PDF file.

-        Authors should provide a clear explanation on the experimental error of the proposed research work. In particular, reproducibility of the phenomena described in the manuscript should be clearly stated in the “Results and Discussion” section; besides, some notes in the “Materials and Methods” section should be added highlighting which kind of experimental approach has been followed to check the reproducibility of the proposed system, the latter being of noteworthy importance in the present research field.

-        References: an article submitted to a journal should be consistent with the contents that it typically proposes in its table of contents. However, by checking the references of this manuscript, I did not find any articles published in this journal: this sounds rather strange. Maybe, Authors could check better the topics recently addressed by this journal, studying its table of contents and enriching the Introduction (as mentioned above) with some articles connected to this field.

Author Response

(The authors gave the same response as above.)

Round 2

Reviewer 1 Report

The manuscript has been revised well and can be accepted by the journal.

Reviewer 2 Report

Approved without corrections

Reviewer 3 Report

The manuscript has been properly amended and I recommend its publication.